# Explaining the difference between men's and women's football

**Luca Pappalardo** [1]*, **Alessio Rossi** [2], **Michela Natilli** [2], **Paolo Cintia** [2]

**1** Institute of Information Science and Technologies (ISTI), National Research Council (CNR), Pisa, Italy,
**2** Department of Computer Science, University of Pisa, Pisa, Italy

* luca.pappalardo@isti.cnr.it

**Data Availability Statement:** Data were generated by company Wyscout as part of this study and provided to us for research purposes. Data are not publicly available, but aggregated data are available on request by contacting ISTI-CNR (luca.

## Abstract

Women's football is gaining supporters and practitioners worldwide, raising questions about what the differences are with men's football. While the two sports are often compared based on the players' physical attributes, we analyze the spatio-temporal events during matches in the last World Cups to compare male and female teams based on their technical performance. We train an artificial intelligence model to recognize if a team is male or female based on variables that describe a match's playing intensity, accuracy, and performance quality. Our model accurately distinguishes between men's and women's football, revealing crucial technical differences, which we investigate through the extraction of explanations from the classifier's decisions. The differences between men's and women's football are rooted in play accuracy, the recovery time of ball possession, and the players' performance quality. Our methodology may help journalists and fans understand what makes women's football a distinct sport and coaches design tactics tailored to female teams.

## 1 Introduction

Women's football took its first steps thanks to the independent women of the *Kerr Ladies* team, who gave the most significant impetus to this sport since the early twentieth-century [1]. As time passed, the *Kerr Ladies* intrigued the English crowds for their ability to stand up to male teams in numerous charity competitions. The success and enthusiasm of these events aroused concerns within the English Football Association, which on December 5th, 1921, decreed that "football is quite unsuitable for females and ought not to be encouraged", and requested "the clubs belonging to the Association to refuse the use of their grounds for such matches" [1]. Unfortunately, this measure drastically slowed down the development of women's football, which, after a long period of stagnation, resurfaced in the first half of the 1960s in Europe's Nordic countries, such as Norway, Sweden, and Germany. From that moment on, the development of women's football was unstoppable, spreading to the stadiums of Europe and the world and carving out a notable showcase among the most popular sports in the world. From 2012 the number of women academies has doubled [2], with around 40 million girls and women playing football worldwide nowadays [3].

pappalardo@isti.cnr.it) or SoBigData
(info@sobigdata.eu).

**Funding:** LP, AR, MN and PC have been funded by
EU project H2020 SoBigData++ RI, grant #871042.

**Competing interests:** The authors have declared
that no competing interests exist.

In the last decade, the attention around women's football has stimulated the birth of statistical comparisons with men's football [2, 4, 5]. Bradley et al. [6] compare 52 men and 59 women, drawn during a Champions League season, and observe that women cover more distance than men at lower speeds, especially in the final minutes of the first half. However, at higher speed levels, men have better performances throughout the game [6]. Perroni et al. [7] show that speed dribbling with and without the ball is higher in male players than in female ones. Cardoso de Araújo et al. [8] highlight that men show a higher explosive capacity, intermittent endurance, sprint performance, and jump height than women. Moreover, women show lower blood lactate, maximal heart rate, and distance covered during an incremental endurance test than men. Sakamoto et al. [4] examine the shooting performance of 17 men and 17 women belonging to a university league, finding that women have lower average values than men on ball speed, foot speed, and ball-to-foot velocity ratio [4]. Pedersen et al. [3] question the rules and regulations of the game and, taking into account the average height difference between 20–25 years-old men and women, estimate that the "fair" goal height in women's football should be 2.25 m, instead of 2.44 m. Gioldasis et al. [5] recruit 37 male and 27 female players from an amateur youth league and find that, while among male players, there is a significant difference between roles for almost all technical skills, among female players, just the dribbling ability presents a significant difference. Sakellaris [9] finds that, in international football competitions, female teams have a higher average number of goals scored per match than their male counterparts. Finally, Van Lange et al. [2] follow 157 female and 207 male young Dutch footballers to investigate the tendency to stop the game to permit a teammate's or opponent's care on the ground, finding that women show, on average, a greater willingness to help.

An overview of the state of the art cannot avoid noticing that current studies focus on physical features and analyze small samples of male and female players using data collected on purpose. At the same time, although massive digital data about the technical behavior of players are nowadays available at an unprecedented scale and detail [10–15], investigations of the differences between women's and men's football from a technical point of view are still limited. Is the intensity of play in women's matches higher than men's ones? Are women more accurate than men in passing? Furthermore, does the statistical distribution of male players' performance quality differ from that of female players?

This article analyzes a large dataset describing 173k spatio-temporal events that occur during the last men's and women's World Cups: 64 and 44 matches, respectively, and 32 men's and 24 women's teams with 736 male players and 546 female players. To the best of our knowledge, ours is the largest sample of men's and women's football matches and players. We quantify players' and teams' performance in several ways, from the number of game events generated during a match to the proportion of accurate passes, the velocity of the game, the quality of individual performance, and teams' collective behavior. We then tackle the following interesting question: *Can a machine distinguish a male team from a female based on their technical performance only?*

Based on the use of a machine learning classifier, we show that men's and women's football do have apparent differences, which we investigate by extracting global and local explanations from the classifier's decisions. Opening the classifier's black box allows us to reveal that, while the game's intensity is similar, the differences between men's and women's football are rooted in play accuracy, time to recover ball possession, and the typical performance quality of the players.

Our methodology is helpful to several actors in the sports industry. On the one hand, a deeper understanding of female and male performance and playing style differences may help coaches and athletic trainers design training sessions, strategies, and tactics tailored for

women players. On the other hand, our results may help sports journalists tell, and football fans understand what makes women's football a distinct sport.

## 2 Football data

We use data related to the last men's World Cup 2018, describing 101,759 events from 64 matches, 32 national teams, 736 players, and the last women's World Cup 2019, with 71,636 events 44 matches, 24 national teams, and 546 players. Each event records its type (e.g., pass, shot, foul), a time-stamp, the player(s) related to the event, the event's match, and the position on the field, the event subtype, and a list of tags, which enrich the event with additional information [10] (see an example of an event in Table 1). Events are annotated manually from each match's video stream using proprietary software (the tagger) by three operators, one operator per team and one operator acting as responsible supervisor of the output of the whole match. We have recently released the dataset regarding the men's World Cup 2018 [10], in companion with a detailed description of the data format, the data collection procedure, and its reliability [10, 16]. Match event streams are nowadays a standard data format widely used in sports analytic for performance evaluation [11, 13, 16, 17] and advanced tactical analysis [18–20]. Fig 1a shows some events generated by a player in a match. Fig 1b shows the distribution of the total number of events in our dataset: on average, a football match has around 1600 events, whereas a couple of matches have up to 2200 events.

## 3 Technical performance

*Do technical characteristics of men's and women's football significantly differ, statistically speaking?* To answer this question, we define variables that describe relevant technical aspects of the game and show for which of them there is a statistical difference between men and women. In particular, we investigate three technical aspects: *(i)* intensity of play (Section 3.1); *(ii)* shooting distance (Section 3.2); and *(iii)* performance quality (Section 3.3).

### 3.1 Intensity of play

The intensity of play is associated with a team's chance of success [19, 21]. Here, we measure the intensity of play in terms of volume and velocity.

 **3.1.1 Volume.** For each team in a match, we compute the total number of events and the number of specific event types (duels, fouls, free kicks, offside, passes, and shots) [10].

**Table 1. Example of event corresponding to an accurate pass.** eventName indicates the name of the event's type: there are seven types of events (pass, foul, shot, duel, free kick, offside and touch). eventSec is the time when the event occurs (in seconds since the beginning of the current half of the match); playerId is the identifier of the player who generated the event. matchId is the match's identifier. teamId is team's identifier. subEventName indicates the name of the sub-event's type. positions is the event's origin and destination positions. Each position is a pair of coordinates (x, y) in the range [0, 100], indicating the percentage of the field from the perspective of the attacking team. tags is a list of event tags, each describing additional information about the event (e.g., accurate). A thorough description of this data format and its collection procedure can be found in [10].

```
{"eventName": "Pass",
"eventSec": 2.41,
"playerId": 3344,
"matchId": 2576335,
"teamId": 3161,
"positions": [{"x": 49, "y": 50}],
"subEventName": "Simple pass",
"tags": [{"id": 1801}]}
```

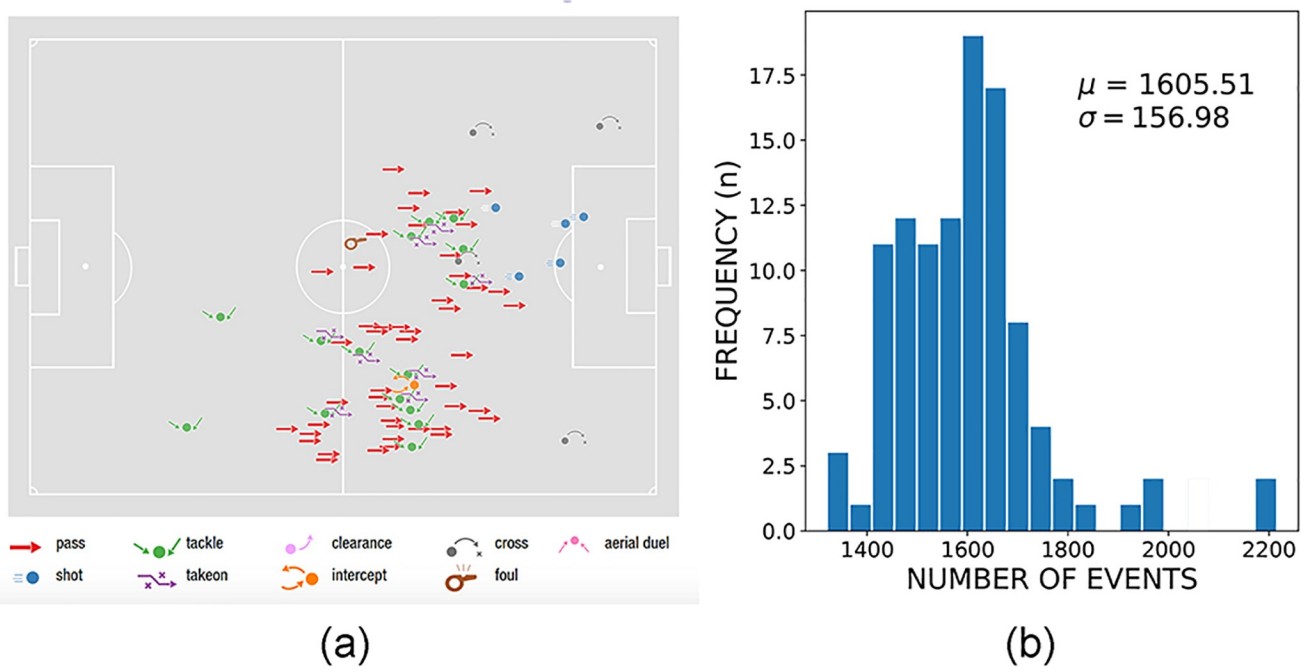

**Fig 1.** (a) Example of events observed for a player in our dataset. Events are shown at the position where they have occurred. This plain "geo-referenced" visualization of events allow understanding how to reconstruct the player's behavior during the match. (b) Distribution of the number of events per match. On average, a football match in our dataset has 1600 events.

Although, on average, men's matches show more events that women's ones, this difference is not statistically significant (unpaired t-score = 1.40, p-value = 0.16, see Table 2). Women's matches have, on average, more free kicks, duels, others on the ball (i.e., accelerations, clearances, and ball touches) and passes but fewer fouls than men's matches (Table 2). Additionally, men's passes are also more accurate than women's ones (unpaired t-score = 8.95, p-value < 0.001). Finally, the number of fouls is lower in woman matches compared to men ones (unpaired t-score = 5.68, p-value < 0.001), resulting in a correct game.

**3.1.2 Velocity.** The average pass velocity PassV($g$) measures the average time between two consecutive passes in a match $g$. The average ball recovery time RecT($g$) measures the average time for a team to recover ball possession in $g$ (Supplementary Information 1 in S1 File). The interruption time StopT($g$) indicates the time spent between two consecutive actions (i.e., time to take a free-kick, a corner kick, or a throw-in). The average pass length PassL($g$) measures the average time between a team's two consecutive shots in a match and the average distance between a pass's starting and ending points, respectively. For all of these features, we perform an unpaired t-test to detect differences between men and women (Table 2). We find that women's PassV($g$) (unpaired t-score = 8.69, p-value < 0.001) is lower than men's one, denoting a higher velocity of passes in men's football (unpaired t-score = 3.540, p-value < 0.001). At the same time, women's RecT($g$) is lower than male's one (unpaired t-score = 5.41, p-value < 0.001), i.e., women regain ball possession faster. In contrast, men's passes PassL($g$) (unpaired t-score = 3.54, p-value < 0.001) are on average larger than women's ones.

## 3.2 Shooting distance

We explore the spatial distribution of the positions where male and female players perform free kicks and shots (Fig 2) and quantify shooting distance ShotD as the Euclidean distance

**Table 2. Statistical difference of technical features between male and female teams.** The summary data for both women and men are report as mean±standard deviation per matches. Grey rows indicates features for which the difference between men and women is statistically significant. The highest values are highlighted in bold.

| Event | Women | Men | t-score | p-value |
|---|---|---|---|---|
| # events | 1522.62±93.82 | 1549.62±99.55 | 1.40 | 0.16 |
| # shots | 21.98±6.03 | 21.52±5.72 | -0.40 | 0.69 |
| # fouls | 19.95±5.94 | **26.94±6.41** | 5.68 | <0.001 |
| # passes | 790.86±98.76 | **861.67±101.25** | 3.57 | 0.001 |
| # free kicks | **102.70±11.85** | 90.05±10.62 | -5.75 | <0.001 |
| # duels | **419.91±53.77** | 394.52±62.25 | -2.18 | 0.03 |
| # offside | **3.88±2.91** | 2.91±1.86 | -2.39 | 0.02 |
| # others | **149.98±26.08** | 141.19±24.65 | -1.76 | 0.05 |
| # accurate passes | 311.66±127.17 | **375.67±138.30** | 3.49 | <0.001 |
| Pass accuracy (AccP) | 0.76±0.08 | **0.84±0.05** | 8.95 | <0.001 |
| Pass velocity (PassV) | 2.83±0.12 | **2.99±0.17** | 8.69 | <0.001 |
| Recovery Time (RecT) | 19.58±10.37 | **27.32±10.14** | 5.41 | <0.001 |
| Stop time (StopT) | 18.92±3.38 | **23.27±2.99** | 6.98 | <0.001 |
| Pass length (PassL) | 19.53±1.53 | **20.32±1.70** | 3.54 | <0.001 |
| Shot distance (ShotD) | 18.39±1.90 | **19.99±1.74** | 4.47 | <0.001 |
| Average PR ($PR_{avg}$) | -0.01±0.01 | **0.01±0.01** | 9.01 | <0.001 |
| Std PR ($PR_{std}$) | 0.05±0.03 | 0.05±0.03 | -0.40 | 0.69 |
| H-indicator (H) | 1.21±0.27 | **1.32±0.36** | 2.49 | 0.01 |
| Flow centrality (FC) | 0.058±0.004 | **0.059±0.003** | 2.11 | 0.04 |

from the position where the shots start to the center of the opponents' goal. To find a statistical difference between men and women, we use the non-parametric Mann-Whitney U-Test. On average, men players kick the ball from a greater distance than women (p-value < 0.001, Table 2).

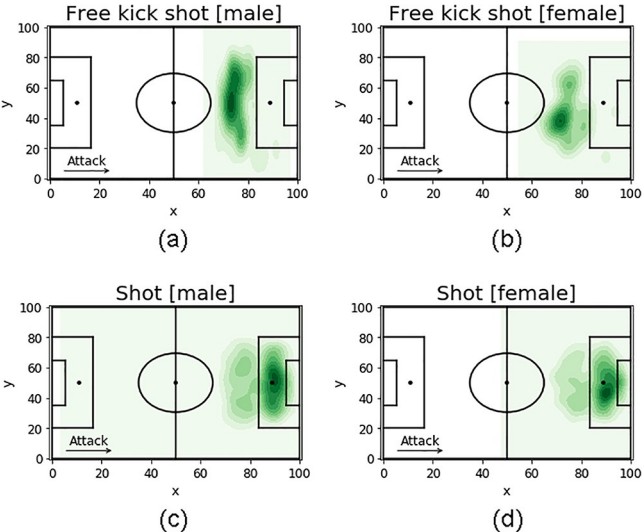

(a)

(b)

(c)

(d)

**Fig 2. Heatmaps describing the pitch zones from where free-kick shots and shots in motion are more likely to be made by male and female players, computed as the kernel estimate of the first grade intensity function, where the event points are the free-kick shots and the shots in motion, and the football field is the region of interest.** The darker is the green, and the higher is the number of free-kick shots and shots in motion in that field zone. The pitch length (x) and width (y) are in the range [0, 100], which indicates the percentage of the field starting from the left corner of the attacking team.

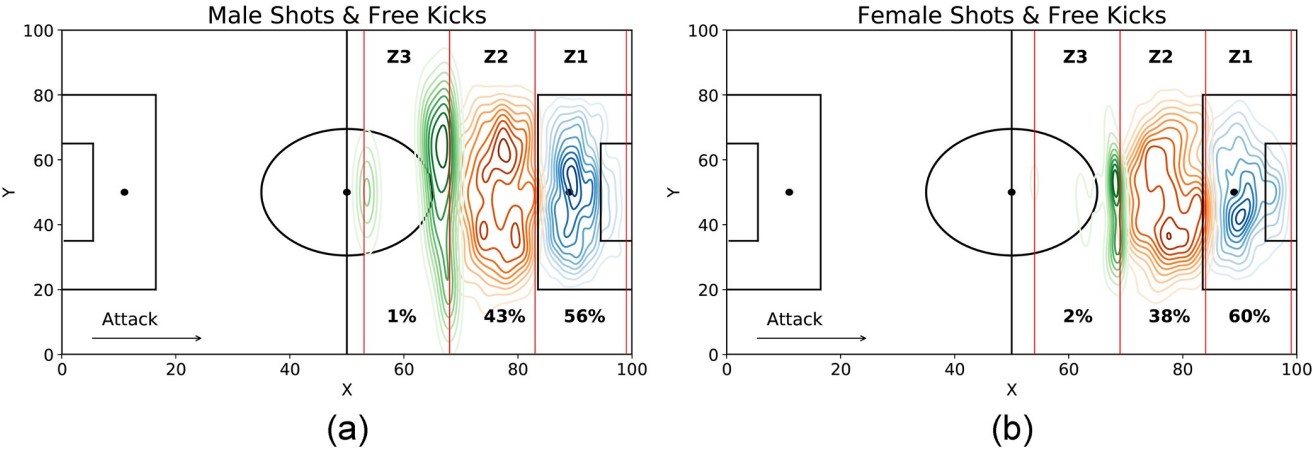

**Fig 3. Pitch zones from where free-kicks and shots in motion are more likely to be made by male players (a) and female players (b).** We split the attacking midfield into three equal zones: Z1 is the area closest to the opponents' goal, Z3 the furthest, and Z2 the zone in the middle. In each zone, we show the percentage of free kicks and shots in motion made in that zone and depict the kernel estimate of the First Grade Intensity function, where the event points are the free-kicks and the shots in motion, and the football field is the region of interest. The darker the color, the higher is the number of events in a specific field position. Female zones are 1.1 meters closer to the opponents' goal than male zones.

To take into account that men and women may have a different perception of distance to the opponents' goal, we split the attacking midfield into three zones $Z1$, $Z2$ and $Z3$, according to the two distributions of shooting distance, i.e., looking at a shot's minimum and the maximum starting positions (Fig 3). $Z1$ is the area closest to the goal, $Z3$ the furthest, $Z2$ the zone in the middle. The zones of women are 1.1 meters closer to the goal than the zones of men (p-value < 0.001).

We then use a z-test for proportions with two independent samples to verify whether there is a difference in the shooting activity between men and women. Female teams have a higher percentage of shots from their Z1 zone than male teams (p-value = 0.01); the opposite is true in the Z2 shooting area (p-value = 0.004). Finally, female teams have a higher percentage of shots from their Z3 shooting area (p-value = 0.02) than male teams.

### 3.3 Performance quality

We use the PlayeRank (PR) algorithm [16] to compute the PR score, which quantifies a player's performance quality in a match (Supplementary Information 2 in S1 File—for details on the algorithm) resuming the players' performance goodness by a data-driven approach. PR score is robust in agreeing with a ranking of players given by professional football scouts, given its capability of describing football performance comprehensively [16]. For each match $g$, and for both teams, we compute the mean and the standard deviation of the individual PR scores, $PR_{avg}(T, g)$ and $PR_{std}(T, g)$, respectively. High values of $PR_{avg}(T, g)$ indicate that the players in team $T$ perform well in match $g$, on average. High values of $PR_{std}(T, g)$ indicate a large variability of PR across the teammates in match $g$. Male players have higher $PR_{avg}$ than females players (unpaired t-score = 9.01, p < 0.001) but similar $PR_{std}$ (unpaired t-score = -0.40, p-value = 0.69). We find statistical difference in the PR score between men and women for left fielders only (Fig 4).

We also explore the differences in the collective behavior of male and female teams computing the passing networks, i.e., graphs in which nodes are players and edges represent passes between teammates in a match [19, 22–25]. From the passing network of a team $T$ in a match $g$

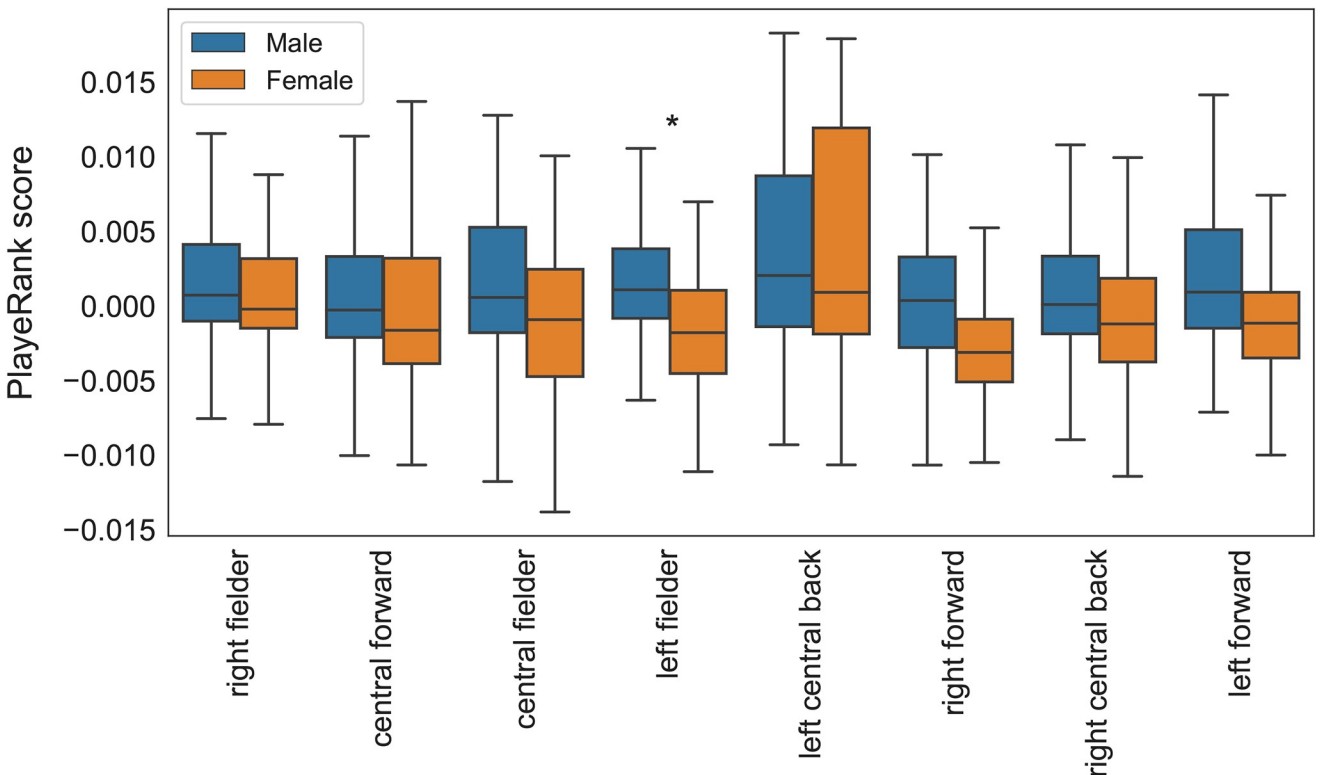

**Fig 4. PlayeRank score by role fro male and female players.** Asterisks indicate significant statistical difference between male and female for that role.

we derive the H indicator $H(T, g)$ [19, 21] and the team flow centrality $FC(T, g)$ [22], two ways of quantifying the goodness of a team's performance in a match [10] (see Supplementary Information 3 in S1 File).

$H(T, g)$ summarizes different aspects of a team's passing behaviour, such as the average amount $\mu_p$ of passes and the variance $\sigma_p$ of the number of passes managed by players [19]. The higher the $\sigma_p$, the higher is the heterogeneity in the volume of passes managed by the players. Differently, a player's flow centrality in a match is defined as their betweenness centrality in the passing network [22]. The team flow centrality, $FC(T, g)$, is hence defined as the average of the flow centralities of players of team $T$ in match $g$ [22].

Table 3 shows the top ten male and female teams with highest average H indicator $H_{avg}$, the average PR score $PR_{avg}(T)$, and average FC score $FC_{avg}(T)$. Spain is the male team with the best overall team performance ($H_{avg}^{(M)}(\text{Spain}) = 1.67$), and so is Japan in the women's World Cup ($H_{avg}^{(F)}(\text{Japan}) = 1.56$). In general, the H indicator of male teams ($H_{avg}^{(M)} = 1.32$) is higher (unpaired t-score = 2.67, $p < 0.02$) than female teams' one ($H_{avg}^{(F)} = 1.21$). Similarly, the FC indicator of male teams ($FC_{avg}^{(M)} = 0.059$) is slightly higher (unpaired t-score = 2.11, $p < 0.04$) than female teams' one ($FC_{avg}^{(W)} = 0.058$).

Since H, FC, and PR quantify different aspects of a team's performance, they are weakly correlated (Table 3). As a matter of fact, no statistical significant correlation was detected among these collective parameters (H vs PR: r = -0.12, $p < 0.01$; H vs FC: r = 0.07, $p < 0.01$; PR vs FC: r = 0.32, $p < 0.05$).

**Table 3. Average H indicator, FC, and PR score for each team in the two competitions.**

| Team | Sex | $H_{avg}$ | Team | Sex | $PR_{avg}$ | Team | Sex | $FC_{avg}$ |
|---|---|---|---|---|---|---|---|---|
| Spain | M | 1.670 | USA | F | 0.077 | Mexico | M | 0.064 |
| Egypt | M | 1.604 | France | F | 0.039 | Argentina | M | 0.063 |
| Denmark | M | 1.592 | Belgium | M | 0.038 | Germany | M | 0.063 |
| Japan | F | 1.565 | Germany | F | 0.037 | Spain | M | 0.063 |
| Australia | M | 1.538 | Australia | F | 0.036 | Morocco | M | 0.063 |
| England | F | 1.530 | Italy | F | 0.035 | Japan | F | 0.062 |
| Chile | F | 1.518 | England | F | 0.034 | England | F | 0.062 |
| Iran | M | 1.470 | Russia | M | 0.034 | Canada | F | 0.062 |
| England | M | 1.447 | Croatia | M | 0.034 | USA | F | 0.062 |
| Tunisia | M | 1.445 | Sweden | F | 0.034 | Scotland | F | 0.061 |
| Peru | M | 1.422 | Netherlands | F | 0.033 | Sweden | F | 0.061 |
| Serbia | M | 1.420 | England | M | 0.031 | Brazil | M | 0.061 |
| Scotland | F | 1.405 | Scotland | F | 0.030 | Korea | M | 0.061 |
| Poland | M | 1.374 | Brazil | F | 0.030 | Peru | M | 0.061 |
| Uruguay | M | 1.367 | France | M | 0.030 | Belgium | M | 0.060 |
| Colombia | M | 1.363 | Brazil | M | 0.029 | Netherlands | F | 0.060 |
| Belgium | M | 1.355 | Tunisia | M | 0.029 | Portugal | M | 0.060 |
| Sweden | M | 1.348 | Portugal | M | 0.027 | Poland | M | 0.060 |
| South Africa | F | 1.344 | Japan | M | 0.027 | Switzerland | M | 0.060 |
| Panama | M | 1.336 | Spain | M | 0.026 | Costa Rica | M | 0.060 |
| Russia | M | 1.327 | Argentina | M | 0.026 | Japan | M | 0.060 |
| Argentina | M | 1.318 | Colombia | M | 0.026 | Germany | F | 0.060 |
| Costa Rica | M | 1.306 | Norway | F | 0.023 | Colombia | M | 0.060 |
| Korea | M | 1.301 | Switzerland | M | 0.023 | Tunisia | M | 0.059 |
| Iceland | M | 1.291 | Uruguay | M | 0.022 | Brazil | F | 0.059 |
| Jamaica | F | 1.284 | Nigeria | M | 0.020 | Saudi Arabia | M | 0.059 |
| Brazil | M | 1.276 | Sweden | M | 0.019 | France | F | 0.059 |
| Netherlands | F | 1.272 | Senegal | M | 0.019 | Norway | F | 0.059 |
| Portugal | M | 1.261 | Canada | F | 0.019 | Spain | F | 0.059 |
| Norway | F | 1.258 | Spain | F | 0.018 | Iran | M | 0.059 |
| Nigeria | F | 1.251 | Korea | M | 0.018 | Australia | M | 0.058 |
| France | M | 1.241 | Japan | F | 0.016 | Nigeria | M | 0.058 |
| Thailand | F | 1.234 | Denmark | M | 0.015 | Australia | F | 0.058 |
| New Zealand | F | 1.232 | Mexico | M | 0.015 | Iceland | M | 0.058 |
| Croatia | M | 1.210 | Cameroon | F | 0.015 | Serbia | M | 0.057 |
| Japan | M | 1.205 | Iceland | M | 0.014 | Korea | F | 0.057 |
| Saudi Arabia | M | 1.196 | Germany | M | 0.014 | France | M | 0.057 |
| Switzerland | M | 1.194 | Poland | M | 0.014 | Chile | F | 0.057 |
| Canada | F | 1.186 | Serbia | M | 0.014 | England | M | 0.057 |
| Brazil | F | 1.185 | Saudi Arabia | M | 0.014 | Sweden | M | 0.056 |
| Australia | F | 1.177 | Peru | M | 0.013 | Uruguay | M | 0.056 |
| Morocco | M | 1.169 | Argentina | F | 0.013 | Senegal | M | 0.056 |
| Korea | F | 1.156 | Egypt | M | 0.013 | Croatia | M | 0.056 |
| Senegal | M | 1.146 | Morocco | M | 0.013 | Denmark | M | 0.056 |
| USA | F | 1.144 | Australia | M | 0.013 | Egypt | M | 0.056 |
| Mexico | M | 1.130 | Costa Rica | M | 0.009 | China PR | F | 0.055 |
| Sweden | F | 1.128 | Panama | M | 0.008 | New Zealand | F | 0.055 |

*(Continued)*

**Table 3.** (Continued)

| Team | Sex | $H_{avg}$ | Team | Sex | $PR_{avg}$ | Team | Sex | $FC_{avg}$ |
|------|-----|-----------|------|-----|------------|------|-----|------------|
| China PR | F | 1.098 | Chile | F | 0.008 | Nigeria | F | 0.055 |
| France | F | 1.095 | Jamaica | F | 0.008 | Panama | M | 0.055 |
| Argentina | F | 1.080 | Korea | F | 0.008 | Jamaica | F | 0.054 |
| Nigeria | M | 1.075 | Iran | M | 0.008 | Russia | M | 0.054 |
| Germany | F | 1.066 | South Africa | F | 0.008 | Argentina | F | 0.054 |
| Spain | F | 1.055 | Thailand | F | 0.007 | Cameroon | F | 0.053 |
| Italy | F | 1.051 | China PR | F | 0.007 | Italy | F | 0.052 |
| Cameroon | F | 0.943 | Nigeria | F | 0.007 | South Africa | F | 0.052 |
| Germany | M | 0.807 | New Zealand | F | 0.003 | Thailand | F | 0.052 |

## 3.4 In summary

Our statistical analysis reveals that male and female teams do differ in many technical characteristics (Table 2):

- Men and women matches have a similar average number of events and shots;

- Women shows a more loyal game compared to men.

- Men perform more passes per match with a higher accuracy indicating a higher volume of play and a better technical quality of the men compared to woman;

- Men perform longer passes and shoot from a longer distance than women, presumably due to the physical differences between genders (e.g., men have greater strength in the legs, which allows them to shoot from farther away);

- The typical performance quality of male teams, in terms of pass volume, heterogeneity, centrality, and PR score, is higher than women's one. This result could be related to the different playing style;

- Women's ball recovery time is shorter than men's, denoting either a better capability of women to recover the ball or a lower capability to retain it, and characterizing a more fragmented game in women's football. This aspect is affected by the women shows a higher number of duels, free kicks, offside, and others on the ball (e.g., clearances, accelerations, and ball touches).

## 4 Team gender recognition

Having established that women's and men's football differ in many technical characteristics related to the intensity of play, shooting distance, and performance quality, we now turn to the question: *Can we design a machine learning classifier to distinguish between a male and a female football team?* If the classifier can correctly distinguish between male and female teams, this highlights *a fortiori* that men's and women's football significantly differ in their technical characteristics. Machine learning can capture the interplay between technical features, and explanations extracted from the constructed classifier can reveal further insights on the differences between men and women football [26].

As a first step, we describe the behavior of a team *T* in match *g* by a performance vector of 19 variables and associate it with a target variable:

- number of events (# events) and number of events of each type (# shots, # fouls, # passes, # free kicks, # duels, # offside, # others, # accurate passes);

- percentage of accurate passes AccP, average shooting length ShootL and average pass length PassL;

- average time between passes PassV;

- average time to regain ball possession RecT and how long a team takes before re-starting the game through a free-kick, a corner kick or a throw-in StopT;

- the H-indicator H, the team flow centrality FC, the average PR score $PR_{avg}$ and its standard deviation $PR_{std}$.

- the target variable indicates whether the team is male (class 1) or female (class 0).

We build a supervised classifier and use 20% of the dataset to tune its hyper-parameters through a grid search with 5-folds cross-validation. We use the remaining 80% of the dataset to validate the model using leave-one-team-out cross-validation: in turn, we leave out all matches of one team and train the model using all matches of the remaining teams. We assess the performance of the model using four metrics [27]:*(i)* accuracy, the ratio of correct predictions over the total number of predictions;*(ii)* precision, the ratio of correct predictions over the number of predictions for the positive class (male);*(iii)* recall, the ratio of correct predictions over the total number of instances of the positive class (male);*(iv)* F1 score, the harmonic mean of precision and recall.

We try several learners to construct different classifiers (Decision Tree, Logistic Regression, Random Forest, and AdaBoost). All classifiers achieve good performance (see Fig 5), with an average relative improvement of 67% in terms of F1-score over a classifier that always predicts the team's gender randomly (Table 4). The best model, AdaBoost, has an improvement of 93% over the baseline in terms of the F1-score. These results indicate that a classifier can distinguish between male and female teams on the only basis of the performance variables.

Inspecting the reasoning underlying the model's decisions can provide us more profound insights into the differences between men's and women's football. We extract global (i.e., inference based on a complete dataset) and local (i.e., inference about an individual prediction) explanations from the best model (AdaBoost) using SHAP (library released for Python, https://github.com/slundberg/shap), a method to explain each prediction based on the optimal Shapley value [28]. The Shapley value of a performance variable is obtained by composing several variables and average change depending on the presence or absence of the variables to determine the importance of a single variable based on game theory [28]. The interpretation of the Shapley value for variable value $j$ is: the value of the $j$-th variable contributed $\phi_j$ to the prediction of a particular instance compared to the average prediction for the dataset [30].

Fig 6 shows the global explanation of AdaBoost, in which variables are ranked based on their overall importance to the model in accordance with shap values. Pass accuracy (AccP) is way far the most important feature to classify a team's gender. Recovery time (RecT), average interruption time (StopT), pass velocity (PassV), pass length (PassL), # duels and # passes, $PR_{avg}$, # fouls and $PR_{std}$ are other essential features for the decision making process.

Fig 7 shows the summary plot that combines feature importance and feature effects, where each point indicates a team. The position of a feature on the y-axis indicates the importance of that feature to the model's decision. A point's color, in a gradient from blue (low) to red (high), indicates its numerical value. A point on the x-axis indicates the associated shap value: positive values indicate that a team is more likely to be male; negative values that it is more likely to be female. Higher values of PassAcc (red points) are associated with higher shap

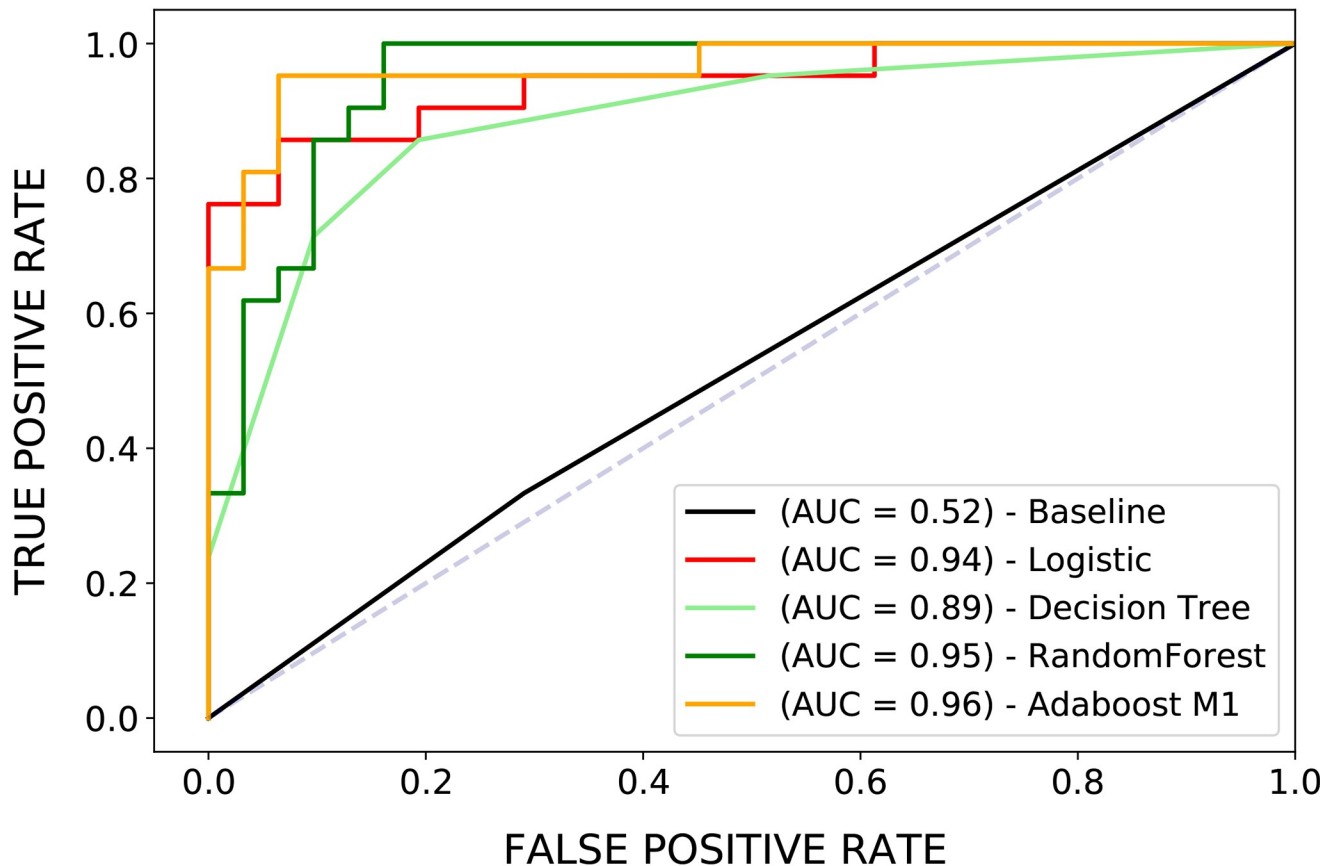

**Fig 5. ROC curves for the implemented classifiers.** They trace the true positive rate and the false positive rate as the probability threshold changes, i.e., the threshold beyond which an observation is assigned to class 1 (male team). When the true positive rate and the false positive rate are 0, the threshold is 1 (all the observations are classified as class 0) [29]. In this case, the true positive rate is the percentage of male teams correctly classified, and the false positive rate is the percentage of female teams mistaken as male, using a given threshold. The actual thresholds are not shown. The AUC represents the area under the curve; the larger the AUC, the better the classifier [29]. Random Forest and Adaboost M1 show the best predictive performance.

values. This indicates that male players are typically more accurate in passing, a property used by the classifier to discriminate a male team from a female one. Similarly, high values of RecT are associated with a higher probability of a team to be male, highlighting *a fortiori* that a more fragmented play characterizes female teams.

Fig 8a refers to the men's World cup 2018, Croatia vs. France. AdaBoost correctly predicts that France is a male team, basing its decision on five main variables: $PR_{avg}$, #passes, AccP, PassV, and RecT. France has RecT(France, CRO vs FRA) = 38.24, #passes(France, CRO vs

**Table 4. Table of the leave-one-team out cross-validation results (i.e., accuracy, precision, recall and F1-score) computed on the training dataset of each machine learning classifiers used to predict a football team in a game as male (*class 0*) or female (*class 1*).** The baseline classifier always predicts by respecting the training set's class distribution, which is balanced. The percentages in the table refer to the improvement of machine learning model compared to the baseline results.

| Classifier | Accuracy | Precision | Recall | F1-Score |
|---|---|---|---|---|
| AdaBoost.M1 | **0.93 (93%)** | **0.80 (70%)** | **0.92 (119%)** | **0.85 (93%)** |
| Random Forest | 0.86 (46%) | 0.69 (45%) | 0.82 (95%) | 0.73 (65%) |
| Decision Tree | 0.85 (77%) | 0.68 (44%) | 0.79 (88%) | 0.71 (61%) |
| Logistic | 0.79 (64%) | 0.64 (36%) | 0.79 (88%) | 0.66 (50%) |
| Baseline | 0.48 | 0.47 | 0.42 | 0.44 |

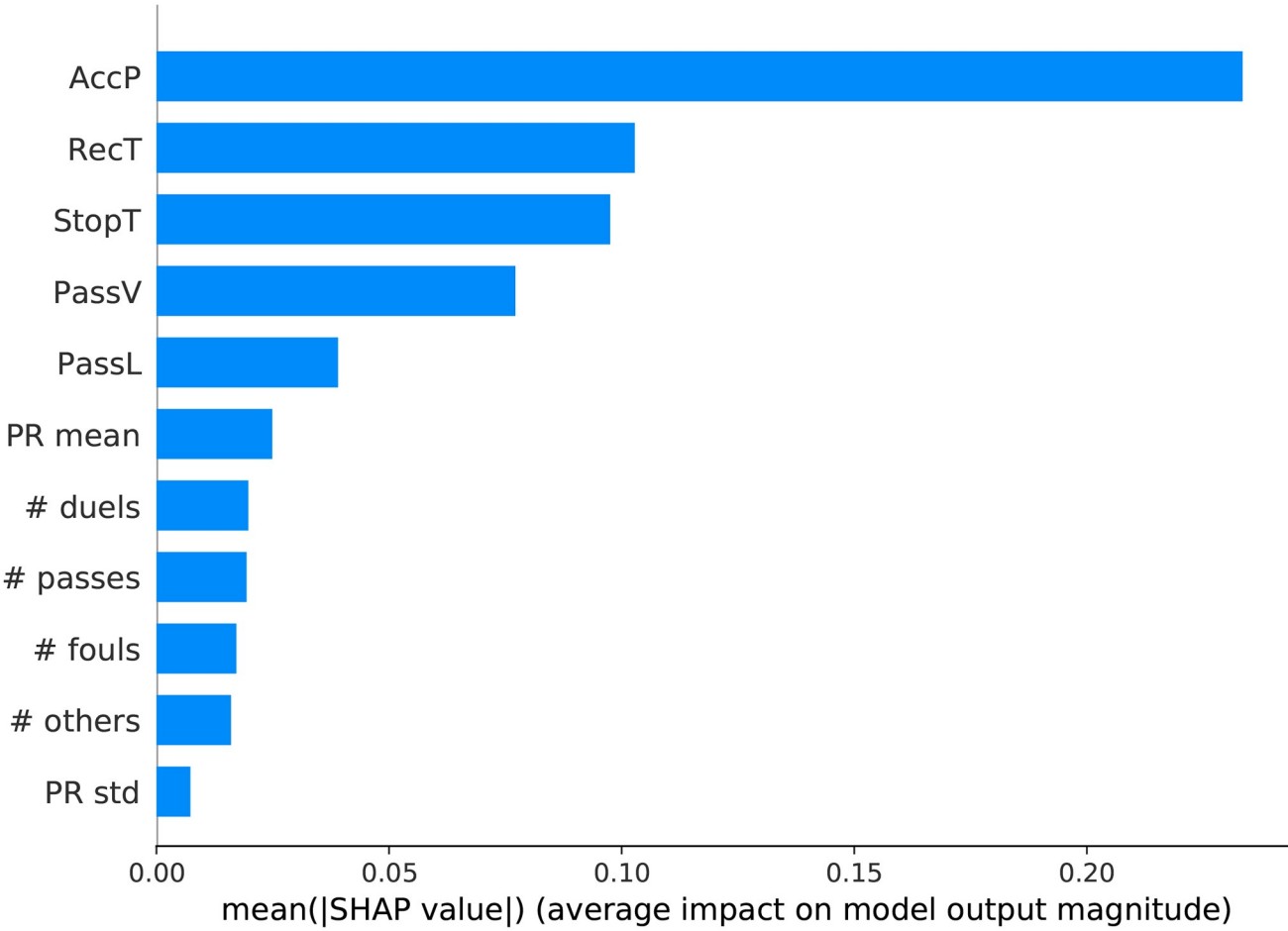

**Fig 6. Ranking of features importance (mean Shap value) extracted from the team gender classifier.**

FRA) = 241 and $PR_{avg}$(France, CRO vs FRA) = 0.05, closer to the typical values of men's football ($RecT^{(M)}$ = 27.32, #passes$^{(M)}$ = 394.43, $PR_{avg}^{(M)}$ = 0.01) than to those of female's football ($RecT^{(F)}$ = 19.58, passes$^{(F)}$ = 430.84, $PR_{avg}^{(F)}$ = −0.01). In contrast, AccP(France, CRO vs FRA) = 0.77 and PassV(France, CRO vs FRA) = 2.68, which are closer to the typical values of a female team (AccP$^{(F)}$ = 0.76, PassV$^{(F)}$ = 2.83, Table 2) than to those of a male one (AccP$^{(M)}$ = 0.84, PassV$^{(M)}$ = 2.99, Table 2). Overall, the sum of the shap values indicates that France played a match in accordance with the typical characteristics of a male team.

Fig 8b shows the prediction of a match in the women's World Cup 2019, USA vs. Spain. In this case, AdaBoost correctly predicts that USA is a female team, basing its decision mainly on AccP, $PR_{std}$, StopT, RecT, and PassV. USA has RecT(USA, USA vs SPA) = 28.94 and StopT (USA, USA vs SPA) = 30.34, closer to the typical values of men's football ($RecT^{(M)}$ = 27.32 and StopT$^{(M)}$ = 23.27, Table 2) than to those a women's football ($RecT^{(M)}$ = 19.58 and StopT$^{(M)}$ = 18.92, Table 2). In contrast, the values of AccP(USA, USA vs SPA) = 0.81 and PassV(USA, USA vs SPA) = 2.83, more similar to those of women teams (Table 2). Overall, the sum of the shap values leads the model to classify US as a female team.

Fig 9a and 9b visualize the predictions of the AdaBoost classifier on a test set of 31 men's matches and 21 women's matches concerning the two most important variables, AccP and RecT. In just two cases out of 21, AdaBoost misclassifies a female team as a male one (Fig 9b).

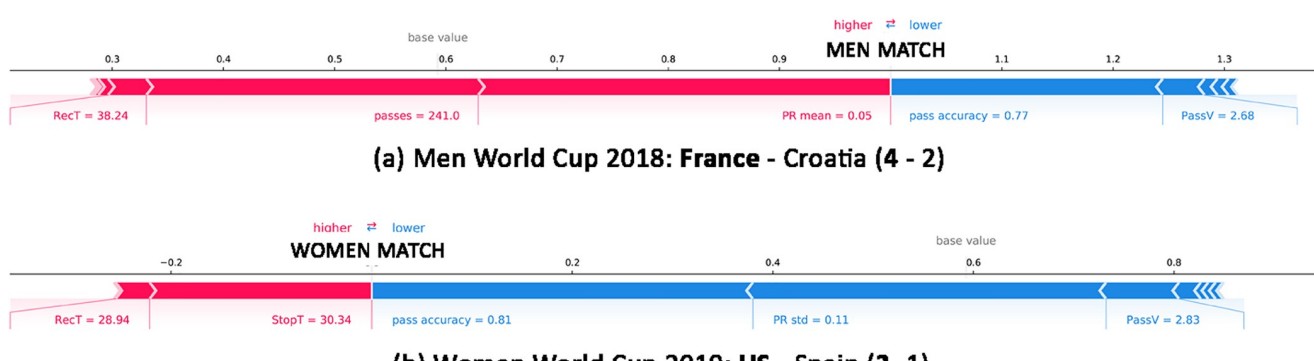

**Fig 7. Distribution of the impact of each feature on the team gender classifier.** The color represents the feature value (red high, blue low); and position of the point indicates the Shap value.

**(a) Men World Cup 2018: France - Croatia (4 - 2)**

**(b) Women World Cup 2019: US - Spain (2 -1)**

**Fig 8. Local Shap explanations for two examples in our dataset: France in France vs Croatia and USA in match USA vs Spain.** Feature values that increase the probability of a team to be male are shown in red, those decreasing the probability are in blue.

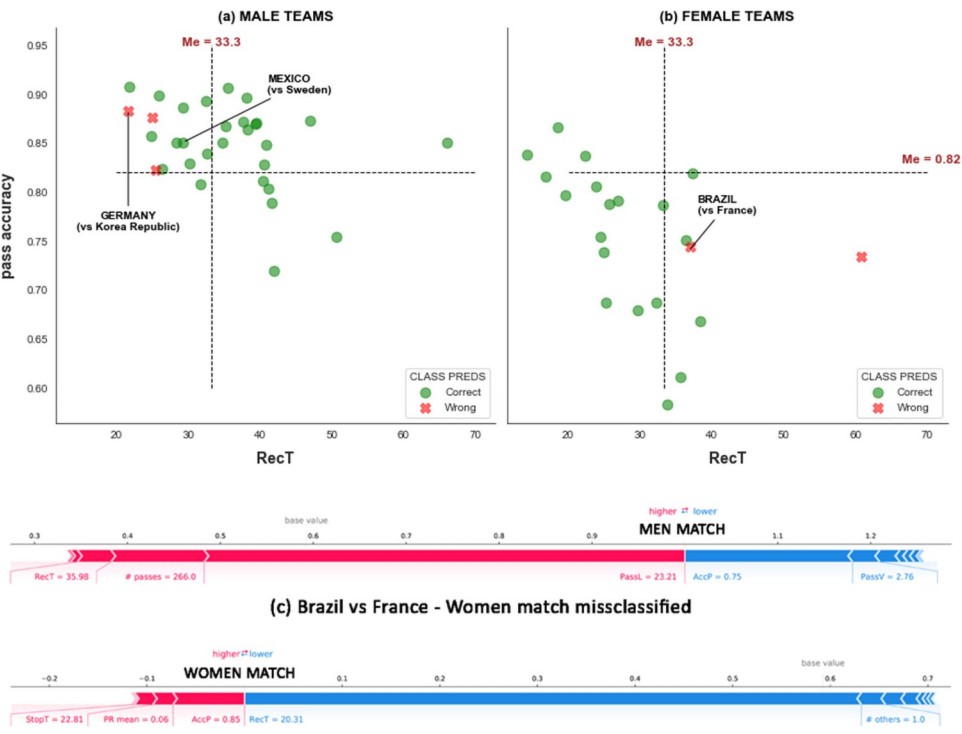

**Fig 9.** (a, b) Scatter plots displaying AccP versus RecT for a test set of teams in male matches (a) and female matches (b). Circles indicate a team correctly classified by the team gender classifier; crosses indicate a mistake by the classifier. The dashed lines are at the median values for the two variables over the entire data set. In plots (c) and (d), we report the local Shap explanations of two misclassified examples.

For example, in Brazil vs. France of the women's World Cup, RecT(Brazil, BRA vs. FRA) = 35.89 and AccP(Brazil, BRA vs. FRA) = 0.75 (Fig 9c), which leads the model to misclassify it as a male team because those values are more typical of women's football than of men's football.

In just three cases out of 31, a male team is misclassified as a female (Fig 9a, red crosses). For example, in match Sweden vs Mexico of the men's World Cup, Mexico is correctly classified as a male team: its values of AccP(Mexico, SWE vs MEX) = 0.85 and RecT(Mexico, SWE vs MEX) = 30 are indeed close to the typical values of men's football. In contrast, in match Germany vs. South Korea, Germany is misclassified as a female team, mainly because RecT(Germany, GER vs KOR) = 20.31 makes it more similar to a female team (RecT$^{(F)}$ = 19.58) than to a male one (RecT$^{(M)}$ = 27.32, see Table 2 and Fig 9d).

The misclassified women's teams have on average AccP$^{(F,\text{wrong})}$ = 0.76 > AccP$^{(F)}$ = 0.75, and a RecT$^{(F,\text{wrong})}$ = 31 > RecT$^{(F)}$ = 29. Moreover, on average StopT$^{(F,\text{wrong})}$ = 19, which is greater than StopT$^{(F)}$ = 18 among all female teams. The misclassified male teams have AccP$^{(M,\text{wrong})}$ = 0.81 < AccP$^{(M)}$ = 0.84 (close to AccP$^{(F)}$ = 0.75), and RecT$^{(M,\text{wrong})}$ = 36 < RecT$^{(M)}$ = 37 (RecT$^{(F)}$ = 29). In both cases, AccP and RecT play a fundamental role in confusing the classifier.

## 5 Conclusions

While current works focus on the differences in physical characteristics between men and women, we reconstruct a mosaic of the differences in playing style using spatio-temporal match events related to the last men's and women's World Cups. However, the only

performance metric relevant to the classifier's classification is the PR score, which captures how good players were on average during a match, rather than the team's playing style as captured by FC and the H indicator. Therefore, our model learns how to detect the differences in the performance and the technical characteristics of the teams rather than their playing style.

Our analysis reveal that differences do exist in several technical features: the time between two consecutive events and the time required to recover possession are the lowest in women's football, and women's game is more "loyal", i.e., women do fewer fouls than men). At the same time, men are typically more accurate in passing, and they kick the ball from a greater distance than women. Among the metrics that characterize team performance, just the PlayeRank score [16] reveals significant differences among men's and women's football.

The inspection of a model that classifies team gender from the technical features through local explanations provides a novel perspective to reason about the difference between men and women in football, highlighting the reason behind the peculiar cases in which the classifier has been "fooled" by a team's technical performance.

Our results are open to various interpretations. The lack of statistically significant difference in the number of events and shots suggests that, overall, men's and women's football have similar play intensity. In contrast, the higher accuracy of passes in men's matches may be due to the higher technical level of male players, which may be rooted in the fact that national teams in the men's World Cup are mainly composed of professional players. In contrast, several female national teams (e.g., Italy) are composed of non-professional players or professional players for a short time. This difference reflects in a lower training time spent by women and therefore in a lower technical level compared to men, as previous studies demonstrate that training time is related to technical capabilities [7, 31, 32]. In this regard, dedicating more time to train specific technical capabilities, such as neuromuscular (i.e., strength) and cognitive (i.e., decision-making, visual searching processes, ability to maintain alert) functions, is crucial to make the training of women more effective [7, 33]. Although women's football is progressively shifting to professionalism and technical level is increasing rapidly, there is still a technical gap between the two sports.

The shorter recovery time observed for women's matches may be due to the lower pass accuracy (i.e., more balls lost), to a better capacity to press the opponents and recover ball possession (e.g., high number of duels), and the higher number of interruptions (i.e., offside and free kicks). Differently, player centrality is typically higher in men's football, denoting the presence of "hubs" that centralize the game (higher flow centrality) and higher variability in the performance quality across teammates (higher H indicator and PR score). In other words, passes in women's football are more uniformly distributed across the teammates.

Women's football also has a preference for short passes over long balls. Since accurate long balls are more difficult than short ones, this preference may be a solution to compensate for women players' lower technical level or different physical characteristics. Indeed, several technical variables are linked to the physiological and anthropometric differences between genders: for example, passing and shooting distances are affected by muscle strength and anthropometrical factors, which differ between the sexes [7].

As future work, we plan to investigate differences in men's and women's football in national tournaments and to investigate to what extent these differences vary nation by nation and between national and continental competitions. Is the difference we found in this paper more marked in the longer competitions for clubs?

## Supporting information

**S1 File.**
(PDF)

## Acknowledgments

We thank WyScout Spa for providing the match events, Daniele Fadda for his support on data visualization, and Giuseppe Pontillo for his contribution.

## Author Contributions

**Conceptualization:** Luca Pappalardo, Alessio Rossi, Michela Natilli, Paolo Cintia.

**Data curation:** Luca Pappalardo.

**Funding acquisition:** Luca Pappalardo.

**Investigation:** Luca Pappalardo.

**Methodology:** Luca Pappalardo.

**Project administration:** Luca Pappalardo.

**Supervision:** Luca Pappalardo, Paolo Cintia.

**Validation:** Luca Pappalardo, Alessio Rossi, Michela Natilli.

**Visualization:** Luca Pappalardo, Alessio Rossi.

**Writing – original draft:** Luca Pappalardo, Alessio Rossi, Michela Natilli.

**Writing – review & editing:** Luca Pappalardo, Alessio Rossi, Michela Natilli, Paolo Cintia.

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
