## [Decision Letter · Decision Letter 0]

7 Apr 2021

PONE-D-21-02337

Explaining the difference between men's and women's football

PLOS ONE

Dear Dr. Pappalardo,

Thank you for submitting your manuscript to PLOS ONE. After careful consideration, we feel that it has merit but does not fully meet PLOS ONE’s publication criteria as it currently stands. Therefore, we invite you to submit a revised version of the manuscript that addresses the points raised during the review process.

This is a strong paper that shows that the traditional machine learning classifiers, which you could also argue they are not necessarily state-of-the-art machine learning techniques since they have been around for decades, can identify female and male teams with high accuracy based on the set of input variables specified. As suggested by the reviewers, the paper requires minor revision before it can be accepted for publication. Please refer to the reviewers' comments. 

If you decide to resubmit a revised version, please provide point-to-point responses to each of the comments made by the reviewers. In your response, please explain what revisions have been made to address each of the points raised by the reviewers. If a comment is not addressed, please justify this decision in your response to the reviewer.

We look forward to receiving your revised manuscript.

Kind regards,

Anthony C Constantinou

Academic Editor

PLOS ONE

Journal Requirements:

Reviewers' comments:

Reviewer's Responses to Questions

**Comments to the Author**

1. Is the manuscript technically sound, and do the data support the conclusions?

Reviewer #1: Yes

Reviewer #2: Partly

2. Has the statistical analysis been performed appropriately and rigorously? 

Reviewer #1: I Don't Know

Reviewer #2: Yes

3. Have the authors made all data underlying the findings in their manuscript fully available?

Reviewer #1: Yes

Reviewer #2: Yes

4. Is the manuscript presented in an intelligible fashion and written in standard English?

Reviewer #1: Yes

Reviewer #2: Yes

5. Review Comments to the Author

Reviewer #1: I found this to be a very innovative and informative paper. As the authors note, comparisons of male and female soccer have only recently received empirical attention, and this study provides an informative overview of major differences in technical play, with a focus on spatio-temporal events, along with individual and collective performance. Findings are descriptively strong and make considerable (intuitive) sense and therefore help to build a foundation for a new line of research. I must admit that I do not have the expertise to evaluate the statistical analyses, so I hope that this is covered by other reviewers. But focusing on my expertise and what I can evaluate, I believe this paper is a very strong one.

Some suggestions for revisions are the following:

(1) I found the paper a real pleasure to read. At the same time, the paper’s readability can be improved if the authors use labels that help readers to immediately grasp the the meaning. For example, the indices of H, PR, and FC are not linked to any meaning, it seems. Why not use meaningful labels, which the authors do do for FC (Flow centrality). Also, H and FC are two indicators of “collective” performance. It was not clear to me whether they were correlated. For example, when I look at Table 3, it seems that H and FC are not strongly correlated, because there are not so many overlapping countries in the top 10.

(2) One sizeable differences between men and women is the # of fouls (and more free kicks among women than men). The authors do pay much attention to it, but this seems quite interesting. This allows for at least a bit more interpretation.

(3) The conclusions are straightforward. But I wonder whether it is desirable to provide a bit more discussion to the major findings, linking to the broader literature. What comes to mind is a body of literature examining the role differences in biological make-up between men and women or the role of “cognition” (e.g., executive functioning) that might be relevant to understanding performance in soccer (see research by Lot Verburgh et al., 2014, PlosOne).

(4) The paper needs to be checked on typos (e.g., length rather than length, fro rather than for, etc). Also, I noted that the Dutch name “Paul A.M. Van Lange” should be read as “Van Lange” (not “Lange”) both in the text and references.

Again, very nice paper.

Paul Van Lange

Reviewer #2: Review of manuscript PONE-D-21-02337

The manuscript reports a study attempting to identify variables that can distinguish between men’s and women’s football (soccer). The aim was to train an AI-model so that it can recognize whether a team is male or female. The results reveal differences between men’s and women’s football on several variables and the model, based on computed performance indicators from selected variables, was able to correctly identify a team’s sex in around nine out of ten cases.

I will not claim to be an expert on the specific methods used for building the model, so my comments are related to the choice of variables, the interpretation of the results, and the conclusions. In particular, I am concerned with the validity of the results, and consequently their usefulness.

I find the study interesting, especially the first part attempting to identify variables that differ between men’s and women’s football. I would have liked to see more of this information in the paper and not only as supplementary information (for example the heatmaps showing areas where free-kicks and shots were taken), and I would have liked to see more discussion related to these differences and their possible consequences.

According to the authors, “current studies focus on the physical features” (line 42), while their variables measure technical performance. However, some of the variables may be different between sexes due to other factors than technical abilities, and they may not be so one-dimensional as they may seem. In fact, there are several possible confounding variables that could compromise the interpretation of the results.

I will give a few examples below:

• Several of the variables that are defined as technical, are in fact highly dependent on physiological factors. For example, passing length, as well as shooting distance require (leg-) muscle strength and are also dependent on biomechanical factors that are different between the sexes, notably torques. Thus, the differences may be rather obvious, and they may not reveal any important information about technical performance.

• There are trade-offs between tactical- and technical variables such as for example pass length and pass accuracy. A team that plays longer passes may well use this as a strategy against teams that are more passing-oriented, or as a general strategy if the players are not so technically fluent. Also, a team that plays out from the back would generate plenty more passes, and also higher passing accuracy due to many passes being less risky, whereas a team that more often played out long from the goalkeeper would generate longer passes on average, and at the same time increase the risk, thus decrease the accuracy. Hence, the variable may be contaminated by differences in playing styles, regardless of sex.

• The average pass length, as well as the average shot distance, is not very much shorter in women (1 m, and 1.5 m, respectively) with considerable overlap as is evident from the standard deviations. This means that, although there is an average difference between the sexes, there are so large within-sex variations that the variable is rather poor at distinguishing between teams. For illustration, in the FIFA WC 2018, according to fbref.com the average shot distance varied between 22 m (Saudi Arabia) and 15 m (Serbia). 15 of the male teams had average shot distances below the female average (18.39 m, according to the present manuscript). In the Women’s FIFA WC 2019, the average shot distance varied between 25 m (Argentina) and 15 m (the Champions, USA). Six female teams had average shot distances above the male average of 19.99 m.

My biggest concern is with the validity of the results, thus what can be concluded from them, apart from the fact that it is possible, in most cases to identify a team by its sex. Exactly what is the model identifying? I am not completely convinced that it is performance quality, and that it is performance quality that differs between sexes.

The algorithm was generally able to categorize matches and teams by their sex (how were the matches selected; randomly?). However it made errors in around ten percent of cases.

A model is only as good as it’s variables, and variables that are used to conclude about differences in quality of performance, would need to be validated against actual performance, which is what we can deduct from Table 3. None of the three indicators seem particularly sensitive to sex differences, with both male and female teams among the top scorers. I would have liked to see the complete rankings, which I suspect may lend some explanation to the fact that the model sometimes mischaracterizes teams as belonging to the opposite sex.

For illustration, passing accuracy for the male teams in the most recent FIFA World Cup (still according to fbref.com), varied between 89% (Spain) and 60% (Iran), with the World Champions, France, at around 80%, and the runners up, Croatia, at 78%. In the women’s WC, passing accuracy varied between 83% (Japan) and 59% (Cameroon), with both the Champions, USA, and Netherlands, the runners up, at around 77%. It should also be noted that these numbers are dependent on the relative percentage of short, medium, and long passes for which the long passes in particular were less accurate.

More importantly, however, is the fact that the scores on the indicators used in the model are not very different between male and female teams (lines 139-171). Albeit the average may be different, there is considerable overlap. Furthermore, most of the highest scoring teams on the three variables were not performing particularly well in the tournament. Only one of the quarter-finalists (QF) for each sex (England, in both cases) were among the top ten scorers on the Havg, whereas two female QFs (USA and England), and no male QFs were among the top ten scorers on the FCavg. Some of the top scoring teams were even among the poorest performers in the tournament. It should be conceded that the PRavg, had five female QFs (USA, France, Germany, Sweden, England) among the top ten scorers, along with three male QFs (Belgium, Croatia, Russia), however PR-scores in general were not different between men and women (Fig. 2).

Thus, the model may not recognize male or female teams, but instead teams playing with a certain style. Furthermore, that playing style, whether by a male or a female team, may not be a valid indicator of the quality of performance. For these reasons, I would urge the authors to be much more prudent in their interpretation of their results, and in particular, their conclusions.

I apologize in advance for any errors, and trust that the authors notify the editor in such cases.

6. PLOS authors have the option to publish the peer review history of their article (what does this mean?). If published, this will include your full peer review and any attached files.

Reviewer #1: **Yes: **Paul Van Lange

Reviewer #2: No

---

## [Author Response · Author response to Decision Letter 0]

12 May 2021

see attached file entitled "Revision R1.pdf"

---

## [Decision Letter · Decision Letter 1]

16 Jul 2021

Explaining the difference between men's and women's football

PONE-D-21-02337R1

Dear Dr. Pappalardo,

We’re pleased to inform you that your manuscript has been judged scientifically suitable for publication and will be formally accepted for publication once it meets all outstanding technical requirements.

Kind regards,

Anthony C Constantinou

Academic Editor

PLOS ONE

Additional Editor Comments (optional):

Reviewers' comments:

Reviewer's Responses to Questions

**Comments to the Author**

1. If the authors have adequately addressed your comments raised in a previous round of review and you feel that this manuscript is now acceptable for publication, you may indicate that here to bypass the “Comments to the Author” section, enter your conflict of interest statement in the “Confidential to Editor” section, and submit your "Accept" recommendation.

Reviewer #1: All comments have been addressed

Reviewer #2: (No Response)

2. Is the manuscript technically sound, and do the data support the conclusions?

Reviewer #1: Yes

Reviewer #2: Yes

3. Has the statistical analysis been performed appropriately and rigorously? 

Reviewer #1: Yes

Reviewer #2: Yes

4. Have the authors made all data underlying the findings in their manuscript fully available?

Reviewer #1: Yes

Reviewer #2: No

5. Is the manuscript presented in an intelligible fashion and written in standard English?

Reviewer #1: Yes

Reviewer #2: Yes

6. Review Comments to the Author

Reviewer #1: I very much appreciate the revisions and the explanations of the revisions. What was already a very strong paper -- with very informative and novel data -- has become even stronger. I believe the paper is ready to be published.

Reviewer #2: Review of revised manuscript PONE-D-21-02337R1

Meanwhile, I have read the revised manuscript, and find it much improved. The authors have done a good job with the revision and have amended many of the issues identified by both reviewers.

I appreciate the inclusion of the heatmaps and the figures depicting the distribution of actions across pitch zones, also that many statements have been moderated and the interpretations of the findings are generally more cautious.

I am still not convinced that the algorithm distinguishes between male and female teams based on performance quality, as is stated. This is due to issues regarding the variables included in the algorithm, mentioned in my previous review.

Regardless, I find the study interesting, and the overall discussion is relevant and interesting. Given that the procedures are better explained, and the variables better justified, together with the generally more modest conclusions, I believe that this discussion could now be moved to the general scientific community, as the paper would be publishable in its present form.

7. PLOS authors have the option to publish the peer review history of their article (what does this mean?). If published, this will include your full peer review and any attached files.

Reviewer #1: No

Reviewer #2: No

---

## [Editor Report · Acceptance letter]

26 Jul 2021

PONE-D-21-02337R1 

Explaining the difference between men’s and women’s football 

Dear Dr. Pappalardo:

I'm pleased to inform you that your manuscript has been deemed suitable for publication in PLOS ONE. Congratulations! Your manuscript is now with our production department. 

Kind regards, 

on behalf of

Dr. Anthony C Constantinou 

Academic Editor

PLOS ONE